# Torch-Uncertainty: A Deep Learning Framework for Uncertainty Quantification

**Adrien Lafage,**[*] **Olivier Laurent,**[*] **Firas Gabetni,**[*] **and Gianni Franchi**
U2IS, ENSTA, Institut Polytechnique de Paris

[*] equal contribution

## Abstract

Deep Neural Networks (DNNs) have demonstrated remarkable performance across various domains, including computer vision and natural language processing. However, they often struggle to accurately quantify the uncertainty of their predictions, limiting their broader adoption in critical real-world applications. Uncertainty Quantification (UQ) for Deep Learning seeks to address this challenge by providing methods to improve the reliability of uncertainty estimates. Although numerous techniques have been proposed, a unified tool offering a seamless workflow to evaluate and integrate these methods remains lacking. To bridge this gap, we introduce `Torch-Uncertainty`, a `PyTorch` and `Lightning`-based framework designed to streamline DNN training and evaluation with UQ techniques and metrics. In this paper, we outline the foundational principles of our library and present comprehensive experimental results that benchmark a diverse set of UQ methods across classification, segmentation, and regression tasks. Our library is available at https://github.com/ENSTA-U2IS-AI/Torch-Uncertainty.

## 1   Introduction

With the rapid advancement of artificial intelligence, deep learning models have become integral to high-stakes applications such as healthcare [22], autonomous driving [30], and finance [35], where predictions with reliable confidence scores are critical. These domains require accurate predictions and a clear understanding of their uncertainty, especially when decisions must be made under ambiguous or incomplete information. As a result, quantifying uncertainty has emerged as a fundamental requirement for deploying AI systems in real-world, safety-critical environments [2, 65].

Uncertainty Quantification (UQ) offers tools to evaluate the reliability of model outputs, enabling actions such as triggering human intervention, deferring uncertain decisions, or flagging risky predictions. Despite their success in predictive performance, Deep Neural Networks (DNNs) are often poorly calibrated [31, 47], making them ill-suited for deployment in high-stakes environments. For example, overconfident but incorrect predictions could lead to inappropriate treatment in medical imaging and result in unsafe driving decisions in autonomous vehicles.

To help address these challenges, we introduce `Torch-Uncertainty`, an open-source library facilitating the development, training, and evaluation of deep learning models with principled uncertainty estimation. Built on top of `PyTorch` [66] and `Lightning` [23], `Torch-Uncertainty` offers a unified and extensible architecture that significantly reduces the engineering overhead typically associated with implementing uncertainty-aware models and evaluating their performance on the many dimensions of robustness [80], such as out-of-distribution detection, distribution shift, and selective classification. `Torch-Uncertainty` supports a broad spectrum of learning tasks, including classification, regression, semantic segmentation, and pixel-wise regression. This makes

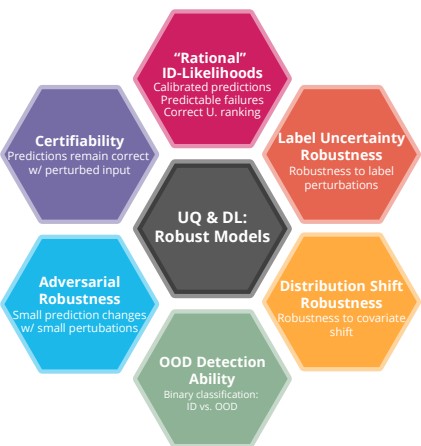

Figure 1: **A suggestion of overview of the many dimensions of robustness and uncertainty quantification in deep learning.** In `Torch-Uncertainty`, we focus on the "rational" in-distribution predictions, distribution-shift robustness and the capacity to detect out-of-distribution samples.

the library a valuable resource for academic research and industrial applications requiring robustness and reliability under distributional shift and noise.

In contrast to existing UQ toolkits [40, 17, 75, 12, 21, 49, 55], `Torch-Uncertainty` distinguishes itself through three key characteristics: **(1) Domain generality:** The library is designed to be highly flexible and applicable to a wide range of data modalities, from mono-modal vision tasks to temporal sequences. **(2) Modular UQ design:** Each uncertainty estimation technique – whether Bayesian, ensemble-based, or deterministic – is implemented modularly, making combining multiple techniques and rapidly prototyping new methods straightforward. We believe this is an essential aspect missing from other libraries, since it involves significant implementation difficulties and requires very high code quality standards. **(3) Evaluation-centric:** Evaluating the robustness of models is key to developing more reliable models: we implement an extensive range of metrics for all tasks, evaluate multiple metrics during validation, and develop advanced, easy-to-use checkpointing methods.

Beyond its core architecture, the library includes several auxiliary components designed to streamline research and development:

- **Uncertainty-aware training routines** based on `Lightning` for efficient experimentation;

- **Standardized evaluation criteria** for comparing UQ methods across tasks and settings;

- **Datasets** on Zenodo, whether official such as MUAD [26] or corrupted for evaluation shift;

- **Pretrained benchmarked model zoo**, hosted on Hugging Face, for plug-and-play testing;

- **Educational resources**, including interactive tutorials, documentation, and use cases aimed at democratizing access to UQ research.

In summary, the main contributions of this paper are as follows:

1. We introduce `Torch-Uncertainty`, the first unified, extensible, domain-general and evaluation-centric `PyTorch`-based library for uncertainty quantification in deep learning.

2. We provide a modular implementation of a wide range of state-of-the-art UQ methods across multiple data modalities and tasks.

3. We benchmark these methods on standard datasets and tasks, offering a reproducible and extensible evaluation framework.

4. We release pretrained models and detailed tutorials to foster adoption by both researchers and practitioners.

## 2 Related Works

**Uncertainty quantification in deep learning**  Deep learning models are affected by multiple sources of uncertainty, which are generally categorized into two main types: *aleatoric uncertainty*, caused by inherent randomness or noise in the data, and *epistemic uncertainty*, arising from limited knowledge about the model parameters or structure [41]. These uncertainties translate into different tasks presented in Figure 1, such as calibration, prediction with rejection (also called selective classification), the detection of out-of-distribution samples using confidence scores or other scores derived from the model predictions, and performance and calibration under distribution shift.

A wide range of techniques has been developed to quantify these uncertainties. While several taxonomies exist, we follow the classification proposed in [29], which organizes uncertainty quantification (UQ) methods into seven broad families: **(1)** Ensemble-based approaches [51, 34] estimate uncertainty by aggregating predictions from multiple DNNs. **(2)** Bayesian approaches [5] explicitly model weight uncertainty using variational inference, stochastic-gradient Markov Chain Monte Carlo, or posterior refinement techniques such as SWAG [59] and TRADI [24]. **(3)** Post-hoc calibration techniques [31, 74] add uncertainty estimation capabilities to pretrained models using lightweight modifications such as temperature scaling, MC Dropout, or Laplace approximation, making them ideal when retraining is costly or infeasible. **(4)** Data augmentation techniques [74] use input perturbations at test time (e.g., test-time augmentation) to derive uncertainty estimates by measuring prediction variability under plausible input transformations. **(5)** Deterministic models for uncertainty estimation [84] produce analytic (closed-form) predictive distributions, such as evidential networks or mean-variance output heads, without requiring sampling or ensembling. **(6)** Interval and conformal prediction methods (CP) [70, 3] wrap around base regressors or classifiers to produce prediction intervals or sets with formal coverage guarantees, without modifying the underlying model. They are particularly effective for finite-sample calibration. **(7)** Gaussian-process-based approaches [84] incorporate Gaussian process (GP) priors into deep models, either through deep kernel learning (DKL) [93], or feature-based surrogates (e.g., SNGP [57]), enabling calibrated non-parametric uncertainty estimates. These families provide complementary tools that allow users to quantify and manage model uncertainty depending on the task and deployment constraints.

**UQ deep learning libraries**  Several libraries have been proposed to support UQ in deep learning. Table 1 compares our library and existing toolkits. Many existing libraries focus on a limited subset of the UQ families. For example, `TorchCP` [40] is a powerful library focused on conformal prediction, making it primarily suited for interval-based methods. Similarly, `Fortuna` [17] supports conformal and Bayesian approaches, emphasizing safety and calibration. Similarly, `TorchUQ` [75] is a library for UQ based on `PyTorch`, which focuses mainly on interval-based methods. `MAPIE` [12] is a dedicated library for conformal prediction, but unlike others mentioned previously, it is not based on `PyTorch`.

On the Bayesian front, `BLiTZ` [21] and `Bayesian-Torch` [49] provide implementations of Bayesian neural networks and variational inference techniques. `BLiTZ` focuses on integrating variational layers into PyTorch models with minimal overhead, while Bayesian-Torch includes support for techniques such as Monte Carlo dropout and Bayes by Backprop.

A library relatively close to ours is `Lightning-UQ-Box` [55], which integrates UQ components within the `Lightning` framework. However, its architecture is more rigid, limiting the flexibility to combine multiple uncertainty techniques or to extend to different modalities.

`Uncertainty Toolbox` [9] primarily targets regression tasks and emphasizes deterministic models with uncertainty estimation; however, it is also not built on the `PyTorch` ecosystem. `GPyTorch` [28] is a specialized library designed for scalable Gaussian process models, focusing strongly on Bayesian techniques. `Uncertainty Baselines` [62] offers a broader selection of UQ methods within the `TensorFlow` framework, but currently supports fewer techniques compared to our library, which – moreover – are not integrated. Similarly, `NeuralUQ` [96] is a recent general-purpose UQ library built on TensorFlow, yet it still includes fewer techniques and less modularity than `Torch-Uncertainty`.

In contrast, our library `Torch-Uncertainty` is designed to be comprehensive and modular. It supports all six prominent UQ families, enabling users to combine and benchmark different methods.

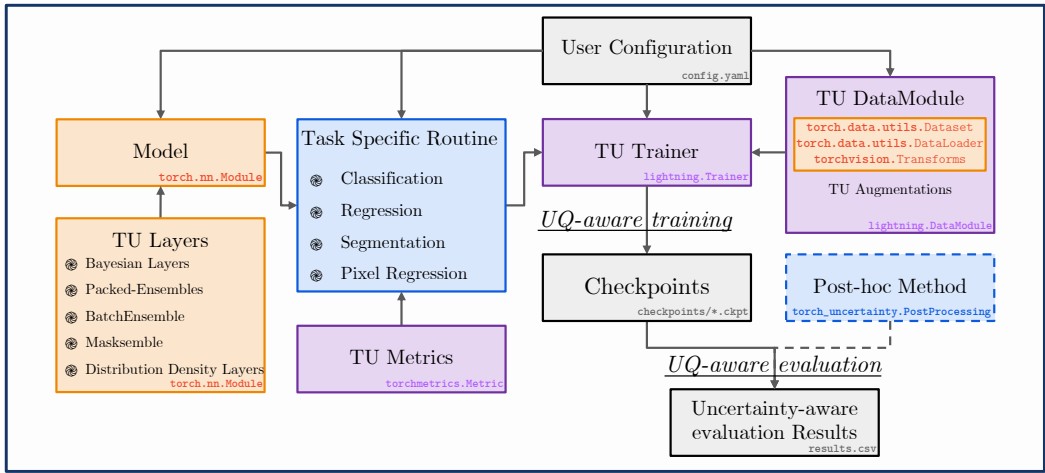

Figure 2: **Overview of Torch-Uncertainty's usage for model training and evaluation.** Post-hoc methods are optional but can improve performance when practitioners can access enough data. UQ and TU stand for uncertainty quantification and `Torch-Uncertainty`, respectively.

## 3 Design and implementation of `Torch-Uncertainty`

`TorchUncertainty` is an open-source framework for uncertainty quantification in deep learning models using `PyTorch`. `Torch-Uncertainty` streamlines the integration of uncertainty-aware methodologies into deep learning pipelines. It offers classification, regression, segmentation, and pixel regression tools, while including pre-implemented uncertainty methods, metrics, and post-processing techniques.

### 3.1 Architecture overview

**Project vision** `Torch-Uncertainty` is designed to be easily extended by external collaborators. We hope the community will take ownership of the library and contribute new methods and applications. To support contributions from a broad range of users, we have established clear contribution guidelines and created a dedicated Discord server (see Appendix H for details). The source code is released under the Apache-2.0 license to encourage widespread use and sharing. To maintain high code quality, Torch-Uncertainty uses `ruff` for compliance with Python coding standards and includes unit tests powered by `pytest` to minimize bugs. As of release v0.7.0, the library achieves nearly complete unit test coverage (around 98%) and integrates automatic tutorials as part of its testing process.

**Global design** Our package benefits from `Lightning` for automated training and evaluation of `PyTorch` models. We define task-specific routines that bridge the gap between `PyTorch` models and `Lightning`'s trainer. These routines define metrics relevant for the task and consider different performance dimensions, i.e., metrics for assessing prediction accuracy, uncertainty estimate quality, out-of-distribution detection, etc. We strictly limit the number of routines to the number of supported tasks to minimize the maintenance burden. Moreover, `Torch-Uncertainty` provides utilities to help implement specific methods, i.e., for instance, `torch_uncertainty.models.wrappers` gives access to model wrappers for smooth ensembling or MC-Dropout and `torch_uncertainty.layers` defines layers for Packed-Ensembles and Bayesian Neural Networks. Figure 2 illustrates the general architecture of our framework.

**TU Routines** Routines in `Torch-Uncertainty` serve as the core building blocks for training and evaluating models with uncertainty quantification in mind. They define standardized frameworks for processing models across the following tasks: classification, regression, pixel-wise regression, and segmentation. Specifically, the routines handle:

- **Task-specific Configurations**: Each type of routine (e.g., `ClassificationRoutine`, `RegressionRoutine`) includes task-adapted functionalities. Specifically, the `RegressionRoutine` can handle models producing distribution parameters, while the `SegmentationRoutine` subsamples pixels to compute metrics efficiently.

- **Training and Evaluation Processes**: They streamline the setup of training loops with integrated uncertainty-aware metrics during validation, enabling **UQ-aware training** as it saves the best checkpoints according to validation metrics. For instance, the `ClassificationRoutine` includes the Accuracy, the Expected Calibration Error, the Negative Log-Likelihood, and the Brier-score. These metrics are tracked and logged throughout the training to provide uncertainty quantification quality insights about the model.

- **Uncertainty Metric Computation**: Routines provide built-in mechanisms to compute different categories of metrics at test time, such as calibration and out-of-distribution detection metrics.

- **Post-processing and Augmentations**: They incorporate post-processing methods like temperature scaling and mixup augmentations, enhancing model performance and reliability.

- **Automated Visualization**: The routines have a parameter to control the generation of plots related to the task at hand (e.g., comparison between predicted and target segmentation masks) or uncertainty quantification (e.g., reliability diagrams). We detail them in Appendix F.

This modular and extensible design enables users to leverage uncertainty quantification techniques effortlessly in deep learning workflows. The following code illustrates how to leverage the `ClassificationRoutine` given a classification model, and some dataloaders (`train_dataloader`, `val_dataloader`, `test_dataloader`).

```python
trainer = Trainer()  # lightning.pytorch.Trainer
cls_routine = ClassificationRoutine(
    model=model,  # torch.nn.Module
    loss=torch.nn.CrossEntropyLoss(),
    optim_recipe=torch.optim.SGD(model.parameters()),
)
# Train and validate the model
trainer.fit(cls_routine, train_dataloader, val_dataloader)
# Evaluate the model
trainer.test(cls_routine, test_dataloader)
```

**Composability of uncertainty quantification techniques** A key design principle of our library is its unified training and inference routine, which enables seamless integration and combination of different UQ techniques. Since all methods are implemented within a common task-specific routine, users can effortlessly compose multiple techniques to create novel hybrid approaches. For example, it is straightforward to construct an ensemble of Laplace approximations, or even an ensemble of Monte Carlo dropout models. These combinations are made possible by simply choosing the appropriate layers for a model, and applying the relevant set of model transformations, or post-processing on the model (See Appendix G for some examples).

### 3.2 Supported uncertainty quantification methods

Among the previously defined seven distinct families, `Torch-Uncertainty` has support for six categories as depicted in Table 1. We chose not to focus on Gaussian Processes as they are challenging to scale to larger models [44]. Most of our implemented techniques are ensembles, Bayesian neural networks, and post-hoc methods, representing the main UQ techniques applied to DNNs. A specificity of `Torch-Uncertainty` is the possibility to choose easily an out-of-distribution (OOD) detection criterion among various choices, such as: maximum class probability, maximum class logit, or entropy. We invite the reader to refer to the documentation of `Torch-Uncertainty` to have a comprehensive list of these criteria.

Table 1: **Uncertainty quantification methods as implemented in the relevant libraries** (✓: implemented): `Torch-Uncertainty` implements and *integrates* a large number of classic methods.

| Category | Method | Torch-Uncertainty | Lightning-UQ-Box | BLiTZ | GPyTorch | TorchCP | Bayesian-Torch |
|---|---|---|---|---|---|---|---|
| *Deterministic* | Deep Evidential [1] | ✓ | ✓ | – | – | – | – |
| | Mean-Variance Est. [64] | – | ✓ | – | – | – | – |
| | Beta-Gaussian Reg. [73] | ✓ | – | – | – | – | – |
| *Ensembles* | Deep Ensembles [51] | ✓ | ✓ | – | – | – | – |
| | BatchEnsemble [90] | ✓ | – | – | – | – | – |
| | Masksembles [20] | ✓ | ✓ | – | – | – | – |
| | MIMO [34] | ✓ | – | – | – | – | – |
| | Packed-Ensembles [52] | ✓ | – | – | – | – | – |
| | Snapshot Ensemble [39] | ✓ | – | – | – | – | – |
| *Bayesian NNs* | Variational BNN [5] | ✓ | ✓ | ✓ | – | – | ✓ |
| | LP-BNN [25] | ✓ | – | ✓ | – | – | ✓ |
| | SWA [43] | ✓ | ✓ | – | – | – | – |
| | SWAG [59] | ✓ | ✓ | – | – | – | – |
| | SGLD [78] | ✓ | ✓ | – | – | – | – |
| | SGHMC [8] | ✓ | – | – | – | – | – |
| *GP-based* | Deterministic UQ [84] | – | ✓ | – | – | – | – |
| | SNGP [57] | – | ✓ | – | – | – | – |
| | Exact / Additive GPs [91] | – | – | – | ✓ | – | – |
| | Variational GP [92] | – | – | – | ✓ | – | – |
| *Quantile / CP* | Conformal Reg. [48] | – | ✓ | – | – | ✓ | – |
| | Conformal Cls. [70, 3] | ✓ | ✓ | – | – | ✓ | – |
| *Post-hoc Methods* | Temperature scaling [31] | ✓ | ✓ | – | – | – | – |
| | Test-Time Aug. [74] | ✓ | ✓ | – | – | – | – |
| | Laplace Approx. [69] | ✓ | ✓ | – | – | – | – |
| | MC-Dropout [27] | ✓ | ✓ | – | – | – | – |
| | MCBatchNorm [79] | ✓ | – | – | – | – | – |
| *OOD Evaluation* | 15 different methods | ✓ | – | – | – | – | – |
| *Diffusion* | CARD [32] | – | ✓ | – | – | – | – |

## 3.3 Supported metrics

Torch-Uncertainty offers by far the widest native metric coverage among the surveyed PyTorch-based uncertainty-focused libraries. It implements 26 distinct metrics spanning over seven task categories: classification, out-of-distribution detection, selective classification, calibration, diversity, regression/depth prediction, and segmentation, so no external code is needed to obtain comprehensive quantitative insights. Additionnally, we provide efficiency-related metrics: the number of parameters and the number of floating point operations. In contrast, `Lightning-UQ-Box`, provides only nine metrics divided into four categories. In comparison, all remaining libraries expose three metrics or fewer and touch at most a single category. More details on supported metrics can be found in the Appendix C.

## 3.4 Supported datasets and applications

Torch-Uncertainty supports multiple applications (classification, regression, segmentation, pixel-level regression) and includes a variety of popular, ready-to-use datasets. As summarized in the Appendix D, Torch-Uncertainty is the only uncertainty-quantification library that ships with a comprehensive, multidomain benchmark suite right out of the box:

- **Corrupted vision datasets**: `Torch-Uncertainty` includes 12 variants ranging from MNIST-C and CIFAR10/100-C,H,N to large-scale ImageNet-A/C/O/R and TinyImageNet-C. We fix, generate, and release corrupted versions of datasets on `Torch-Uncertainty`'s Hugging Face.

- **OOD vision:** We include six popular out-of-distribution sets: Places365, Textures, SVHN, iNaturalist, NINCO, SSB-hard, and OpenImages-O.

- **Dense prediction**: Our framework leverages three semantic-segmentation sets: CamVid, Cityscapes, MUAD; and four depth/texture or synthetic image collections: Fractals, Frost, KITTI-Depth, and NYUv2.

- **UCI tabular data**: The library includes five classical classification sets: BankMarketing, Dota2, HTRU2, OnlineShoppers, SpamBase, and a unified UCI-Regression loader that transparently cycles through 9 regression datasets.

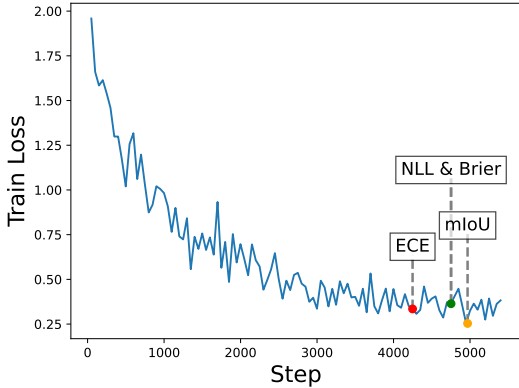

Figure 3: **Best checkpoint positions according to validation metrics.** The model is a UNet optimized on MUAD's semantic segmentation dataset.

All these datamodules use the same `PyTorch/Lightning` API for automatic download, on-the-fly corruptions, balanced splits, and standard normalization. Splits for the corrupted vision, shifted vision, tabular, and dense-prediction sets are handled entirely by Torch-Uncertainty. For the OOD-vision datasets, we follow the exact val/test splits defined in the OpenOOD library – no extra setup is needed

In contrast, `Lightning-UQ-Box` only includes synthetic toy generators, useful for didactic purposes, but insufficient for large-scale comparative analysis. `BLiTZ`, `GPyTorch`, `TorchCP`, and `Bayesian-Torch` do not provide any datasets, leaving data preparation to non-UQ centered libraries.

By tying together a wide range of UQ methods with 27 plug-and-play datasets from image classification and segmentation to depth regression, and tabular tasks – Torch-Uncertainty offers the most complete all-in-one environment for reproducible, cross-domain experiments. This lowers entry barriers and allows researchers to focus on methodological advances rather than dataset plumbing.

### 3.5   Uncertainty-aware training of deep neural networks

An important feature of `Torch-Uncertainty`'s routines is using several callbacks to save the best model for multiple validation metrics. This functionality mitigates the issue that the best model on a given metric might be suboptimal for others. In Figure 3, we report an example showcasing at what epoch the model reaches its best performance on a specific validation metric. By including UQ metrics in the validation procedure, we allow users to more comprehensively track the quality of their models throughout training and automatically store their respective best checkpoint. We also provide a Python object `CompoundCheckpoint` to efficiently save the best checkpoint according to a combination of validation metrics, which would help explore the tradeoff between metrics.

## 4   Benchmarking and experimental evaluation

In this section, we demonstrate the capability of `Torch-Uncertainty` for benchmarking UQ methods on specific tasks. We invite the reader to refer to Appendix A for both experiments' implementation and training details. We report two benchmarks: one on image classification and the other on semantic segmentation. In addition to the task-specific metrics, we evaluate the calibration of the models and their performance in selective classification and out-of-distribution detection. In Appendix E and Appendix I, we reports benchmarks on regression and time-series classification respectively.

The model's calibration is evaluated using the Expected Calibration Error (**ECE**) and the Adaptive Calibration Error (**aECE**). For the selective classification, we consider the Area Under the Risk-Coverage curve (**AURC**), the Area Under the Generalized Risk-Coverage curve (**AUGRC**), the Coverage at 5 Risk (**Cov@5Risk**), and the Risk at 80 Coverage (**Risk@80Cov**). The quality of OOD detection is assessed using the Area Under the Receiver Operating Characteristic Curve (**AUROC**), the Area Under the Precision-Recall curve (**AUPR**), and the False Positive Rate at $95\%$ Recall (**FPR**$_{95}$).

Table 2: **ViT-B-16 benchmark: Classification, Calibration, and Selective Classification.**

| Method | *Classification* | | | *Calibration* | | | *Selective Classification* | | |
|---|---|---|---|---|---|---|---|---|---|
| | Acc (%) | Brier | NLL | ECE (%) | aECE (%) | AUGRC (%) | AURC (%) | Cov@5Risk (%) | Risk@80Cov (%) |
| Single Model | 80.67 | 0.27 | 0.71 | **0.01** | **0.01** | 3.89 | 5 | 64.15 | 9.81 |
| + Temperature Scaling | 80.67 | 0.27 | 0.71 | **0.01** | **0.01** | 3.88 | 4.99 | 64.2 | 9.79 |
| + TTA | 75.12 | 0.49 | - | 0.24 | 0.24 | 11.81 | 21.89 | - | 25.41 |
| Deep Ensemble | **82.19** | **0.25** | **0.65** | 0.03 | 0.03 | 3.46 | 4.44 | 67.58 | 8.54 |
| + Temperature Scaling | **82.19** | **0.25** | **0.65** | 0.01 | 0.01 | **3.44** | **4.41** | **67.92** | **8.49** |
| Packed Ensemble | 79.23 | 0.29 | 0.78 | **0.01** | **0.01** | 4.26 | 5.48 | 62.01 | 10.88 |
| MiMo | 80.59 | 0.27 | 0.72 | 0.02 | 0.02 | 3.8 | 4.84 | 65.67 | 9.63 |

Table 3: **ViT-B-16 benchmark: NearOOD and FarOOD performance.**

| Method | *FarOOD Average* | | | *NearOOD Average* | | |
|---|---|---|---|---|---|---|
| | AUROC (%) ↑ | FPR$_{95}$ (%) ↓ | AUPR (%) ↑ | AUROC (%) ↑ | FPR$_{95}$ (%) ↓ | AUPR (%) ↑ |
| Single Model | 90.75 | 37.92 | 70.33 | 77.96 | 63.08 | 55.29 |
| + Temperature Scaling | 90.44 | 38.62 | 69.59 | 77.72 | 63.45 | 54.93 |
| Deep Ensemble | **92.05** | **33.05** | **72.9** | **78.76** | **61.69** | **56.14** |
| + Temperature scaling | 91.18 | 35.71 | 70.57 | 77.99 | 62.87 | 54.98 |
| Packed ensemble | 89.84 | 38.6 | 66.99 | 76.38 | 65.59 | 52.64 |
| MiMo | 89.13 | 42.34 | 66.65 | 78.05 | 62.84 | 55.67 |

## 4.1 Classification benchmarks

We benchmark the ViT-B/16 architecture across various uncertainty quantification methods, evaluation metrics, and datasets available within `Torch-Uncertainty`. To enhance robustness and support ensemble-based UQ methods, we repeat the entire training pipeline three times with different random seeds, thereby producing a deep ensemble of ViT-B/16 models. All trained weights are made publicly available via the `Torch-Uncertainty`'s Hugging Face repository.

The training process follows a two-stage procedure inspired by the original ViT framework [19]: we first pre-train the model on the large-scale ImageNet-21k dataset [68], and subsequently fine-tune it on the standard ImageNet-1k benchmark [16].

**Benchmarked uncertainty quantification techniques.** In our study, we benchmark a standard ViT-B/16 model with and without Temperature scaling as baselines and ensembles including Deep Ensembles and Packed Ensembles, which aggregate predictions from multiple independently trained models.

**Results Analysis.** Tables 2 and 3 summarize the performance of different ViT-B-16 variants. The *Deep Ensemble* [51] achieves the best overall accuracy (82.19% vs. 80.67% for the single model) and lowest Brier/NLL (0.25/0.65). After temperature scaling, its calibration further improves (ECE=0.01%) with a slight gain in selective classification (*AURC*=4.41%, *Cov@5Risk*=67.9%, *Risk@80Cov*=8.49%). Compact ensemble variants, such as *Packed Ensemble* [52] and *MiMo* [34], reach comparable accuracies (79.2–80.59%) but with weaker performance on other metrics (*Risk@80Cov*=10.9% and 9.6%, respectively). In OOD detection, the *Deep Ensemble* [51] again leads with the highest *FarOOD AUROC* (92.05%) and lowest *FPR$_{95}$* (33.05%), outperforming the single model by +1.3 AUROC and −4.9 FPR. On *NearOOD*, it attains 78.8% AUROC and 61.7% FPR, while *MiMo* [34] remains competitive (78.1% / 62.8%). Overall, ensembles trained independently provide the best accuracy, calibration, and OOD robustness, while compact variants trade some of these gains for efficiency.

## 4.2 Segmentation benchmarks

To demonstrate the capabilities of `Torch-Uncertainty` for benchmarking deep learning approaches, we conduct segmentation experiments using the `SegmentationRoutine`. Based on a UNet architecture [71], we evaluate ensemble approaches on the MUAD dataset [26], whose official implementation is hosted directly in `Torch-Uncertainty`. MUAD contains 3420 image samples for training, 492 for validation, 551 for in-distribution, and 1668 for out-of-distribution test data.

We report the performance of a vanilla UNet model without additional tweaking as baseline, a Monte Carlo (MC) Dropout [27] UNet, one of the simplest UQ baselines, and ensembles including Deep

Table 4: **Semantic segmentation and calibration quality comparison (averaged over three runs) on MUAD using UNet backbones.** All ensembles have 4 subnetworks. We highlight the best performance in bold. Deep Ensembles performs best except for calibration due to augmentations [52].

| | Method | Segmentation | | | | | Calibration (%) ↓ | |
| | | mIoU (%) ↑ | mAcc (%) ↑ | pixAcc (%) ↑ | Brier ↓ | NLL ↓ | ECE | aECE |
|---|---|---|---|---|---|---|---|---|
| | Baseline | 71.55 | 87.65 | 93.59 | 0.10 | 0.18 | 0.51 | **0.42** |
| | + MC Dropout | 68.80 | 85.99 | 92.62 | 0.11 | 0.21 | 1.52 | 2.04 |
| Ensembles | MIMO ($\rho = 0.5$) | 70.95 | 87.32 | 93.15 | 0.10 | 0.20 | **0.44** | 1.04 |
| | BatchEnsemble | 64.88 | 80.78 | 92.06 | 0.12 | 0.24 | 2.73 | 3.18 |
| | Masksemble | 67.62 | 83.14 | 92.87 | 0.11 | 0.21 | 2.08 | 2.61 |
| | Packed-Ensembles | 71.87 | 86.77 | 93.65 | 0.10 | 0.19 | 1.91 | 2.54 |
| | Deep Ensembles | **74.93** | **88.86** | **94.29** | **0.09** | **0.17** | 1.58 | 2.14 |

Table 5: **Selective classification and out-of-distribution detection performance comparison (averaged over three runs) on MUAD using UNet backbones.** All ensembles have 4 subnetworks. We highlight the best performance in bold. For most metrics, Deep Ensembles performs best.

| | Method | Selective Classification (%) | | | | Out-of-Distribution Detection (%) | | |
| | | AURC ↓ | AUGRC ↓ | Cov@5Risk ↑ | Risk@80Cov ↓ | AUPR ↑ | AUROC ↑ | FPR$_{95}$ ↓ |
|---|---|---|---|---|---|---|---|---|
| | Baseline | 0.79 | 0.69 | 96.84 | 1.20 | 18.87 | 81.34 | 57.20 |
| | + MC Dropout | 1.01 | 0.88 | 94.32 | 1.72 | 19.92 | 83.16 | 49.32 |
| Ensembles | MIMO ($\rho = 0.5$) | 0.87 | 0.76 | 95.82 | 1.37 | 18.07 | 80.09 | 59.28 |
| | BatchEnsemble | 1.18 | 1.01 | 92.78 | 2.12 | 19.93 | 83.36 | **48.00** |
| | Masksemble | 0.94 | 0.83 | 94.99 | 1.59 | 20.09 | 83.28 | 48.42 |
| | Packed-Ensembles | 0.77 | 0.68 | 97.02 | 1.19 | 20.40 | 82.56 | 52.97 |
| | Deep Ensembles | **0.63** | **0.56** | **98.53** | **0.93** | **22.45** | **84.03** | 51.40 |

Ensembles (DE) [51] to lighter methods such as MIMO [34], BatchEnsemble [90], Masksemble [20], and Packed-Ensembles (PE) [52]. We consider an ensemble size of 4 for all ensembles, while MC Dropout leverages 10 forward passes.

The methods are evaluated using the built-in metrics of the `SegmentationRoutine`. It includes segmentation-specific metrics: mean Intersection over Union (**mIoU**), the average of the accuracy on each class (**mAcc**), and the accuracy over all pixels (**pixAcc**). Additionally, we report the Brier-score (**Brier**) and the Negative Log-Likelihood (**NLL**) of the target over the categorical distributions predicted by the segmentation model.

In Table 4, we can see that DE, which outperforms other methods on segmentation metrics, is not well calibrated compared to the baseline model. We argue that this result comes from data augmentations at training time, and we emphasize the need for automated calibration quality assessment to detect such behaviors. Concerning selective classification and out-of-distribution detection in Table 5, DE outperforms its counterparts, while PE achieves interesting results with only 25% of the parameters of DE. The performance of PE compared to DE hints that there might not be enough parameters in the subnetworks. Thus, a higher $\alpha$ value (e.g., $\alpha = 3$) would be beneficial. Indeed, $\alpha = 4$ corresponds to PE having the same number of parameters as DE, when the ensemble size is 4. BatchEnsemble has the best **FPR**$_{95}$, but it is also the method with the lowest segmentation capability.

## 5 Conclusion

`Torch-Uncertainty` is a unified, modular, and evaluation-centric library for uncertainty quantification in deep learning. Built on top of `PyTorch` and `Lightning`, it provides a wide range of state-of-the-art UQ techniques implemented across six major methodological families. Our library supports tasks including classification, segmentation, and regression, and comes with pre-trained models, standardized benchmarks, and extensive educational material. As our library is still under development, we invite the community to contribute to this open-source project to create a new standard in UQ for DNNs. Through comprehensive experiments, we demonstrate the utility and extensibility of our framework, paving the way for more robust and uncertainty-aware deep learning models in both academic and industrial contexts.

**Limitations and future directions.** `Torch-Uncertainty`, as opposed to a list of independent methods and scripts, is designed to integrate implemented uncertainty-quantification methods and ease their use and evaluation on sets of tasks and robustness dimensions. This integration significantly increases maintenance and development costs, thereby limiting the library's current comprehensiveness. The authors will strive to continue implementing methods and improving the assessment of their robustness. These difficulties also entail the limitation to the four main tasks presented in the paper and `Torch-Uncertainty`'s specialization on computer vision. The modular structure of the library also makes it more prone to bugs due to compatibility issues between bricks, which we mitigate as much as possible with high code quality standards, unit, and integration tests.

**Societal impact.** Creating an open-source library for uncertainty quantification in Deep Learning can significantly advance research and real-world applications by fostering transparency, reproducibility, and collaboration. It empowers a broader community, including academic researchers, industry practitioners, and developers, to more rigorously assess model confidence and reliability, which is crucial in high-stakes fields like healthcare, and autonomous systems. By democratizing access to state-of-the-art tools, such a library can accelerate innovation, improve model safety, and promote ethical AI deployment across diverse sectors.

## Acknowledgments

The authors thank all the contributors to the library for their help and suggestions, as well as the reviewers for their helpful feedback. The reviewer's comments helped build a roadmap for further improvements to the library.

This work was granted access to the HPC resources of IDRIS under the allocation 2024-AD011014689R1 , 2024-AD011011970R4, and the allocation 2024-AD011015965 made by GENCI. Oliver Laurent acknowledges travel support from ELIAS (GA no 101120237).

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

# Table of Contents

# A   Implementation details

All the hyperparameters used in this paper are available in the configuration files available in the experiments folder of the library.

## A.1   Classification benchmark

Concerning the classification benchmark, we adopted a two-stage training procedure for ViT, following a similar approach to that of [19]. The training procedure for the two stages is detailed in the following paragraphs.

**Stage 1: Pre-training on ImageNet-21k.**   We train ViT-B/16 from scratch on the ImageNet-21k [68] Winter 2021 version, which contains 13,153,500 images across 19,167 classes. Each input image is processed by a transformation pipeline consisting of:

- a random resized crop to $224 \times 224$ with a scale sampled uniformly from $[0.08, 1.0]$,
- a horizontal flip with probability 0.5,
- conversion to tensor,
- channel-wise normalization.

The model is optimized using the AdamW optimizer with the following hyperparameters:

$$\eta_{\max} = 10^{-3}, \quad \text{dropout} = 0.1, \quad \lambda = 0.03, \quad \text{betas} = (0.9, 0.999).$$

The learning rate follows a linear warm-up for the first 10,000 steps, followed by a linear decay schedule. We pre-train for 90 epochs using this configuration.

**Stage 2: Fine-tuning on ImageNet-1k.**   To adapt the model to the ImageNet-1k [16] distribution, we load the pretrained weights, and reinitialize the classification head to accommodate $N = 1000$ target classes.

For the training split, we reuse the pre-training data augmentation pipeline. For validation, images are resized to $256 \times 256$, center-cropped to $224 \times 224$, and normalized. We further split the official validation set into a small validation subset (1%) and a larger test subset (99%) to monitor convergence.

Fine-tuning utilizes stochastic gradient descent (SGD) with a momentum of 0.9, no weight decay, and no dropout. The learning rate is selected from the grid $\{0.003, 0.01, 0.03, 0.06\}$, with a linear warm-up over 500 steps, followed by a cosine decay over 20,000 steps. Training continues until convergence on the validation subset or until reaching the total number of training steps

## A.2   Segmentation benchmark

Concerning the benchmark on the MUAD segmentation dataset containing 15 in-distribution classes and 6 out-of-distribution classes, all models were trained according to the hyperparameters reported in Table 6. During the training stage, we apply the following transformations to the input images:

1. Resizing to evaluation size $(512, 1024)$;
2. Random rescaling with min_scale $= 0.5$ and max_scale $= 2.0$;
3. Random cropping to crop size $(256, 256)$;
4. Random color jitter with brightness $= 0.5$, contrast $= 0.5$ and saturation $= 0.5$;
5. Random horizontal flip with probability $= 0.5$;
6. Channel-wise normalization.

All ensembles use 4 estimators, and MC Dropout leverages 10 forward passes. Packed-Ensembles parameters are $\alpha = 2$ and $\gamma = 1$. We use MIMO with $\rho = 0.5$ and Masksemble with scale $= 2.0$. Concerning BatchEnsemble and Masksemble, the inputs are repeated ($\times 4$) during training.

| Epochs | Batch size | Crop size | Eval size | Optimizer | LR | Weight decay | LR decay | Milestones | Precision |
|--------|-----------|-----------|-----------|-----------|-----|--------------|----------|------------|-----------|
| 100 | 32 | (256, 256) | (512, 1024) | Adam | 1e-2 | 1e-4 | 0.5 | [20,40,60,80] | `bf16-mixed` |

Table 6: Segmentation benchmark hyperparameters

## B  Tutorial details

We provide multiple tutorials accessible in the web documentation of our library, showcasing multiple implemented UQ methods in Torch-Uncertainty. We list below some of the currently available tutorials :

1. **Training a LeNet with Monte-Carlo Dropout**
   In this tutorial, we train a LeNet classifier on the MNIST[54] dataset while keeping `Dropout` active at test time[27]. Multiple stochastic passes yield an empirical posterior, from which both the expected class and predictive variance are extracted.

2. **Improve Top-label Calibration with Temperature Scaling**
   In this tutorial, we use `Torch-Uncertainty` to post-process a pre-trained ResNet-18 (CIFAR-100[50]) with a single learned temperature parameter that rescales logits[31]. The notebook shows how to fit the temperature on a held-out set and achieve a lower Expected Calibration Error (ECE), together with reliability diagrams.

3. **Training a LeNet with Monte-Carlo Batch Normalization**
   This tutorial will apply Monte-Carlo Batch Normalization[79], a post-hoc Bayesian approximation method, to a LeNet with batch normalization layers. Multiple stochastic passes yield an ensemble of logits; TorchUncertainty computes classification, calibration, and selective prediction metrics.

4. **Train a Bayesian Neural Network in Three Minutes**
   In this tutorial, we use `Torch-Uncertainty` to easily train a variational Bayes[5] LeNet with the ELBO loss and to visualize ensemble variance, illustrating epistemic uncertainty.

5. **Deep Evidential Classification on a Toy Example**
   Based on `Torch-Uncertainty`, this tutorial offers an introductory overview of Deep Evidential Classification[1] using a practical example. It tackles the toy problem of classifying MNIST[54] with an MLP whose output is modeled as a Dirichlet distribution. Training minimizes the DEC loss, which combines a Bayesian risk squared-error term with a KL-divergence-based regularizer.

6. **Deep Evidential Regression**
   In this tutorial, we present `Torch-Uncertainty` for Deep Evidential Regression[1] and provide a practical example. We apply DER by tackling the toy problem of fitting $y = x^3$ using a Multi-Layer Perceptron (MLP) neural network model. The output layer of the MLP provides a Normal-Inverse-Gamma distribution, which is used to optimize the model through its negative log-likelihood.

7. **Corrupting Images to Benchmark Robustness**
   This tutorial shows the impact of the different corruption transforms available in the `Torch-Uncertainty` library. Various corruption transforms (noise, blur, weather, JPEG artifacts, . . . ) inspired by ImageNet-C[14] are available. In the tutorial, five severity levels are previewed side by side, allowing users to visualize how data shift tests model robustness.

8. **From a Standard Classifier to a Packed-Ensemble**
   In this tutorial, we demonstrate how to create a Packed-Ensemble[52] starting from the classic CIFAR-10[50] CNN; every `Conv2d` and `Linear` layer is swapped for its Packed version, forming four subnetworks that share computation via grouped convolutions. Better accuracy and uncertainty metrics are achieved with only a modest increase in memory.

9. **Improved Ensemble Parameter-Efficiency with Packed-Ensembles**
   In this tutorial, we train a Packed Ensemble[52] on MNIST[54] and compare it with a deep ensemble[51]. The reported accuracy, Brier score, calibration error, and negative log-likelihood illustrate the efficiency claims made in the Packed-Ensemble paper.

10. **Simple Out-of-Distribution Evaluation**
    This tutorial sets up a CIFAR-100[50] datamodule that automatically provides in-distribution, near-OOD, and far-OOD splits, then runs the `ClassificationRoutine` to collect standard accuracy and OOD metrics: AUROC, AUPR, and FPR95. It also explains how TorchUncertainty integrates the OpenOOD[94] datasets splits by default and how you can plug in your own datasets for custom OOD benchmarking.

11. **Conformal Prediction on CIFAR-10**
    This tutorial introduces Conformal Prediction as a post-hoc method for classification. Using a held-out calibration set, a pretrained ResNet-18 on CIFAR-10[50] is calibrated with three conformal methods (THR, APS, and RAPS)[70, 3]. The notebook measures coverage and set size before and after calibration, visualizes the resulting prediction sets, and even checks their behavior on an OOD dataset (SVHN[63]) to highlight how conformal prediction interacts with distribution shift.

## C  Built-in metrics details

Table 7 compares the metrics supported by six popular PyTorch-based UQ libraries. The metrics are grouped into eight semantic categories, reflecting the most common evaluation axes across classification, regression, and dense-prediction tasks.

`Torch-Uncertainty` stands out by offering the most extensive and diverse support across these categories. It implements a broader set of UQ metrics than existing libraries, addressing various tasks and evaluation dimensions. These metrics encompass performance and uncertainty quantification, all of which are supported natively within the library.

Table 7: **Metrics available in the relevant libraries** grouped by category (✓: implemented)

| Category | Metric | Torch-Uncertainty | Lightning-UQ-Box | BLiTZ | GPyTorch | TorchCP | Bayesian-Torch |
|---|---|---|---|---|---|---|---|
| *Classification* | Accuracy | ✓ | ✓ | ✓ | ✓ | ✓ | ✓ |
| | BrierScore | ✓ | – | – | – | – | – |
| | CategoricalNLL | ✓ | – | – | – | – | – |
| *OOD Detection* | AURC | ✓ | – | – | – | – | – |
| | $FPR_X$ | ✓ | – | – | – | – | – |
| | FPR95 | ✓ | – | – | – | – | – |
| *Selective Classification* | AUGRC | ✓ | – | – | – | – | – |
| | $RiskAt_X Cov$ | ✓ | – | – | – | – | – |
| | RiskAt80Cov | ✓ | – | – | – | – | – |
| | $CovAt_X Risk$ | ✓ | – | – | – | – | – |
| | CovAt5Risk | ✓ | – | – | – | – | – |
| *Calibration* | aECE | ✓ | – | – | – | – | – |
| | ECE | ✓ | ✓ | – | – | – | – |
| | RMSCE | – | ✓ | – | – | – | – |
| | Miscal. Area | – | ✓ | – | – | – | – |
| *Diversity* | Disagreement | ✓ | – | – | – | – | – |
| | Entropy | ✓ | – | – | – | – | ✓ |
| | MutualInformation | ✓ | – | – | – | – | ✓ |
| | VariationRatio | ✓ | – | – | – | – | – |
| *Regression / Depth* | DistributionNLL | ✓ | – | – | – | – | – |
| | Log10 | ✓ | – | – | – | – | – |
| | MAE-Inverse | ✓ | – | – | – | – | – |
| | MAE | ✓ | ✓ | ✓ | – | – | – |
| | MSE | ✓ | ✓ | ✓ | – | – | – |
| | RMSE | ✓ | ✓ | – | – | – | – |
| | MSE-Inverse | ✓ | – | – | – | – | – |
| | MSLE | ✓ | – | – | – | – | – |
| | SILog | ✓ | – | – | – | – | – |
| | ThresholdAccuracy | ✓ | – | – | – | – | – |
| | R2 | – | ✓ | – | – | – | – |
| | SMSE | – | – | – | ✓ | – | – |
| | MSLL | – | – | – | ✓ | – | – |
| | QCE | – | – | – | ✓ | – | – |
| *Segmentation* | Accuracy | ✓ | – | – | – | – | – |
| | mIoU | ✓ | ✓ | – | – | – | – |
| | F1Score | – | ✓ | – | – | – | – |
| *Conformal* | Coverage | ✓ | ✓ | – | – | ✓ | – |
| | Set Size | ✓ | – | – | – | ✓ | – |

# D Datasets details

Appendix D lists the **37** datasets that are built-in `Torch-Uncertainty`. They are grouped by experimental purpose; most modules implement a Lightning-style interface with reproducible splits, standard normalization, and task-specific data augmentations. So, swapping a dataset entails *zero* code changes in the training loop. All competing libraries considered ship *no* datasets.

**Vision:** Core image classification benchmarks spanning low-to-high resolution:

- **MNIST[54]**: 70 000 handwritten digit images ($(28 \times 28)$ px), grayscale, 10 classes.
- **CIFAR-10[50]**: 60 000 color images ($(32 \times 32)$ px) in 10 common object categories.
- **CIFAR-100[50]**: same images as CIFAR-10 but organized into 100 fine-grained classes.
- **TinyImageNet[53]**: 100 000 downsampled ($(64 \times 64)$ px) images across 200 ImageNet classes.
- **ImageNet[16]**: ~1.2 million high-resolution images in 1000 classes for large-scale training.

**Vision - corrupted/shifted:** Robustness stress tests via synthetic corruptions and natural distribution shifts:

- **MNIST-C[61]**: MNIST digits with 15 algorithmic corruptions (e.g., noise, blur).
- **NotMNIST[7]**: Font-based A-J glyphs, mimicking MNIST but with a different style.
- **CIFAR-10/100-C[14]**: CIFAR images under 19 corruption types at five severity levels.
- **CIFAR-10-H[67]**: human annotated "hard" subset of CIFAR-10 for label uncertainty.
- **CIFAR-10/100-N[89]**: CIFAR with naturally noisy labels from real annotators.
- **ImageNet-A[37]**: Adversarial ImageNet examples.
- **ImageNet-C[14]**: ImageNet with the 15 corruption types at five strengths.
- **TinyImageNet-C[14]**: TinyImageNet under the same ImageNet-C[14] corruption types.
- **ImageNet-O[37]**: Out-of-distribution images (100 000 examples) not in ImageNet-1K.
- **ImageNet-R[36]**: Rendition images (art, sketches) of ImageNet classes for style shift.
- **OpenImage-O[88]**: OOD subset drawn from OpenImages.

**Segmentation:** Standard semantic segmentation benchmarks for urban and aerial scenes:

- **CamVid[6]**: Road scene frames ($(360 \times 480)$ px) with 11 semantic classes.
- **Cityscapes[13]**: 5 000 finely annotated street view images in 30 classes.
- **MUAD[26]**: A synthetic dataset for autonomous driving with multiple uncertainty types and tasks.

**Tabular (UCI):** Five classical binary classification tasks with built-in preprocessing:

- **BankMarketing[60]**: customer "yes/no" subscription to a term deposit.
- **DOTA2Games[81]**: match outcome (win/lose) from in-game statistics.
- **HTRU2[58]**: pulsar detection in radio frequency observations.
- **OnlineShoppers[72]**: purchase behavior ("buy" vs. "no buy") from web session logs.
- **SpamBase[38]**: email spam detection (spam vs. non-spam) based on word frequencies.

**Regression:** Ten continuous-target UCI datasets with uniform splits: *Boston[33]*: housing price regression, *Energy[82]*: heating and cooling load prediction for buildings , Naval[11]: submarine propulsion plant state estimation, etc ..

**OOD Eval:** Out-of-distribution image classification datasets with default OpenOOD[94] splits:

- **Places365[95]**: A large-scale scene-recognition dataset with 365 indoor/outdoor classes.

- **Textures[10]**: 5640 texture images organized into 47 perceptual classes

- **SVHN[63]**: Over 600 000 real-world ($32 \times 32$) px RGB digit images cropped from Google Street View.

- **iNaturalist[85]**: A large, fine-grained species-classification dataset with hundreds of thousands of wildlife images (plants, animals, fungi)..

- **NINCO[4]**: Consists of 5879 OOD images across 64 classes, explicitly excluding any ImageNet-1K categories. Designed for rigorous OOD detection evaluation on ImageNet-trained models.

- **SSB-hard[86]**: An out-of-distribution (OOD) dataset for ImageNet-1K classifiers. It defines OOD classes by mining the large ImageNet-21K hierarchy for the 1 000 categories that sit farthest semantically from the original 1 000 training classes, as measured by WordNet similarity.

**Misc. vision:** Auxiliary and multi-modal datasets:

- **Fractals[45]**: procedurally generated fractal images for self supervised pretraining.

- **FrostImages[14]**: synthetically fogged/frosted scenes to study visibility degradation.

- **KITTI-Depth[83]**: stereo and LiDAR-based depth estimation images captured on roads.

- **NYUv2[76]**: aligned RGB and Kinect-derived depth maps of indoor scenes.

Table 8: Datasets / datamodules shipped with each library (✓ = available, - = not supported)

| Category | Dataset / Datamodule | Torch-Unc. | Lightning-UQ-Box | BLiTZ | GPyTorch | TorchCP | Bayesian-Torch |
|---|---|---|---|---|---|---|---|
| *Vision* | Cifar10 | ✓ | - | - | - | - | - |
| | Cifar100 | ✓ | - | - | - | - | - |
| | Mnist | ✓ | - | - | - | - | - |
| | TinyImageNet | ✓ | - | - | - | - | - |
| | ImageNet | ✓ | - | - | - | - | - |
| *Vision (corrupted)* | MNISTC | ✓ | - | - | - | - | - |
| | NotMNIST | ✓ | - | - | - | - | - |
| | CIFAR10C | ✓ | - | - | - | - | - |
| | CIFAR100C | ✓ | - | - | - | - | - |
| | CIFAR10H | ✓ | - | - | - | - | - |
| | CIFAR10N | ✓ | - | - | - | - | - |
| | CIFAR100N | ✓ | - | - | - | - | - |
| | ImageNetA | ✓ | - | - | - | - | - |
| | ImageNetC | ✓ | - | - | - | - | - |
| | ImageNetO | ✓ | - | - | - | - | - |
| | ImageNetR | ✓ | - | - | - | - | - |
| | TinyImageNetC | ✓ | - | - | - | - | - |
| *Vision (shift)* | OpenImageO | ✓ | - | - | - | - | - |
| *Tabular (UCI)* | BankMarketing | ✓ | - | - | - | - | - |
| | DOTA2Games | ✓ | - | - | - | - | - |
| | HTRU2 | ✓ | - | - | - | - | - |
| | OnlineShoppers | ✓ | - | - | - | - | - |
| | SpamBase | ✓ | - | - | - | - | - |
| *Regression* | UCIRegression | ✓ | - | - | - | - | - |
| *Segmentation* | CamVid | ✓ | - | - | - | - | - |
| | Cityscapes | ✓ | - | - | - | - | - |
| | MUAD | ✓ | - | - | - | - | - |
| *OOD* | Places365 | ✓ | - | - | - | - | - |
| | Textures | ✓ | - | - | - | - | - |
| | SVHN | ✓ | - | - | - | - | - |
| | iNaturalist | ✓ | - | - | - | - | - |
| | NINCO | ✓ | - | - | - | - | - |
| | SSB-hard | ✓ | - | - | - | - | - |
| *Misc. vision* | Fractals | ✓ | - | - | - | - | - |
| | FrostImages | ✓ | - | - | - | - | - |
| | KITTIDepth | ✓ | - | - | - | - | - |
| | NYUv2 | ✓ | - | - | - | - | - |

Table 9: **Regression benchmark (averaged over five runs) on UCI Boston & Concrete datasets using an MLP backbone.** All ensembles have 5 subnetworks. We highlight the best performance in **bold**.

| Method | Boston housing | | | | Concrete | | | |
|---|---|---|---|---|---|---|---|---|
| | MAE | RMSE | QCE | NLL | MAE | RMSE | QCE | NLL |
| Baseline | 1.737 | 2.225 | - | - | **3.919** | **5.279** | - | - |
| + Normal | 1.563 | 2.322 | 0.051 | -0.134 | 4.180 | 5.693 | 0.022 | 0.134 |
| + Laplace | 1.598 | 2.322 | 0.036 | -0.113 | 4.154 | 5.906 | 0.036 | 0.180 |
| + Cauchy | 1.688 | 2.485 | 0.037 | 0.022 | 4.367 | 6.257 | 0.041 | 0.310 |
| + Student's T | 1.669 | 2.417 | 0.037 | -0.056 | 4.029 | 5.740 | 0.028 | 0.139 |
| **Bayesian NNs** | | | | | | | | |
| Variational BNN (VI ELBO) | 1.687 | 2.242 | **0.035** | -0.047 | 4.131 | 5.687 | 0.022 | 0.122 |
| **Post-Hoc Methods** | | | | | | | | |
| MC Dropout | 1.706 | 2.262 | 0.079 | -0.074 | 4.459 | 5.948 | 0.048 | 0.251 |
| **Ensembles** | | | | | | | | |
| MIMO ($\rho = 0.5$) | 1.766 | 2.309 | 0.080 | 0.052 | 4.327 | 5.853 | 0.027 | 0.186 |
| BatchEnsemble | 1.799 | 2.358 | 0.080 | 0.038 | 4.312 | 5.881 | 0.032 | 0.200 |
| Packed-Ensembles ($\alpha = 3$) | 1.682 | 2.230 | 0.052 | -0.059 | 4.242 | 5.743 | 0.025 | 0.153 |
| Packed-Ensembles ($\alpha = 4$) | 1.632 | 2.154 | 0.052 | -0.082 | 4.167 | 5.700 | 0.023 | 0.130 |
| Deep Ensembles | **1.509** | **2.040** | 0.071 | **-0.139** | 4.110 | 5.672 | **0.018** | **0.093** |

# E   Regression benchmarks

`Torch-Uncertainty` supports regression tasks, which we showcase with the following benchmark on UCI Regression datasets. We reproduced a simple regression experiment that trains a Multi-Layer Perceptron (MLP). Our benchmark considers the following methods:

- **Baseline** A MLP with 50 hidden neurons and RELU non-linearity, which is the backbone for all our models. It returns a point-wise estimate with no uncertainty estimate whatsoever.
- **Density Network** `Torch-Uncertainty` provides layers (`torch.nn.Module`) to easily estimate distribution parameters. We replace the last layer of our MLP with such layers to estimate Normal, Laplace, Cauchy, and Student's T distributions.
- **Variational BNN** A BNN trained with the ELBO loss, outputting the parameters of a Normal distribution. It uses 10 samples.
- **MC Dropout** At test time, executes 10 forward passes of a Normal Density Network with a dropout rate of 0.1.
- **Ensembles** All ensembles have 5 subnetworks, and produce the parameters of a Normal distribution.

These methods are evaluated on the Mean Absolute Error (**MAE**), the Root Mean Squared Error (**RMSE**), the Quantile Calibration Error (**QCE**), and the Negative Log-Likelihood **rescaled** (**NLL**).

Table 9 reports the performance of the studied methods for the Boston Housing and Concrete datasets. `Torch-Uncertainty` enables the efficient comparison of different distribution families. For instance, in the Boston housing dataset, the Normal distribution appears to be the best, while in the Concrete dataset, it is less clear, as the Student's T distribution has the best **MAE**. Moreover, the Baseline outperforms all other methods that estimate distribution parameters in this dataset. It highlights that simultaneously optimizing the mean and variance might degrade the mean estimation depending on the dataset.

Regarding the ensembles, Deep Ensembles achieve the highest performance, while Packed Ensembles showcase how the number of parameters in the subnetworks affects the results.

# F   `Torch-Uncertainty` **visualization toolbox**

One of the core objectives of `Torch-Uncertainty` is to help practitioners improve the performance of their models while also understanding the limitations of predictive uncertainties. To this end, we

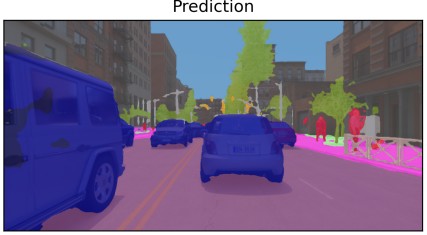
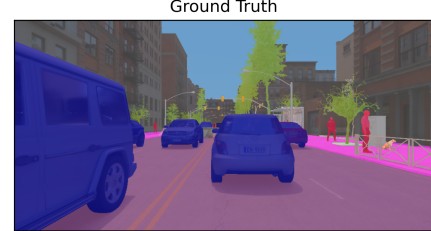

Figure 4: **Example of a prediction visualization available in** `Torch-Uncertainty`**.** The model is a DeepLabV3+ trained on MUAD-Small for 20 epochs.

provide built-in visualizations for the different tasks at hand, ranging from prediction visualization to advanced metric graphs. Notably, for segmentation and image regression, we log prediction images during the training and at test time, as shown in Figure 4. Moreover, we provide detailed plots for calibration and selective classification, available simply with the corresponding `.plot()` methods.

## G   Example of codes

As explained in the main paper, `Torch-Uncertainty` provides routines to simplify the training and the benchmarking of methods on classification, segmentation, and regression tasks (pixel regression is still under heavy development as of `v0.5.0`). Specifically, we focus on designing simple parameters to enable the computation of specific metrics or even apply some post-hoc methods.

**OOD detection in** `ClassificationRoutine` **and** `SegmentationRoutine`**:**  The `eval_ood` parameter indicates that the second dataloader passed to `Torch-Uncertainty`'s `TUTrainer` corresponds to out-of-distribution data. In the code snippet below, we consider an already trained network `model`:

```
trainer = TUTrainer()
cls_routine = ClassificationRoutine(model=model, eval_ood=True)
trainer.test(
    cls_routine, dataloaders=[id_dataloader, ood_dataloader]
)
```

With `eval_ood=True`, the evaluation metrics include the OOD detection metrics: AUROC, AUPR, and FPR95.

**Add a post-hoc method in** `ClassificationRoutine`**:**  In this example, we fit a Temperature Scaler for a model trained on CIFAR-10. `Torch-Uncertainty` provides a `CIFAR10DataModule` class that has a method `postprocess_dataloader()` which is required to fit the post-hoc method:

```
trainer = TUTrainer()
datamodule = CIFAR10DataModule()
cls_routine = ClassificationRoutine(
    model=model, post_processing=TemperatureScaler()
)
trainer.test(cls_routine, datamodule=datamodule)
```

Doing so computes additional metrics evaluating the performance of the post-hoc method on the model.

## H   Contribution protocol

To ease contributing to `Torch-Uncertainty`, we have defined standard guidelines to help with code quality and formatting. Using a specific software development environment, we help any contributor

ensure continuous integration does not break while they implement new features or solve bugs. These guidelines are as follows:

1. Check that `PyTorch` is already installed on your development environment (e.g., conda or venv environments).

2. Clone your personal fork of the `Torch-Uncertainty` repository.

3. Install `Torch-Uncertainty` in editable mode with the development packages.

4. Install pre-commit hooks to guarantee code format and quality when committing, thanks to `ruff`.

We recommend executing the tests locally before pushing on a Pull Request (PR) to avoid multiplying the number of featureless commits. A PR is expected to respect the following conditions:

- The name of the branch is not `main` nor `dev`.

- The PR does not reduce the code coverage of the project.

- The code is documented: the function signatures are typed, and the main functions have clear docstrings.

- The code is mostly original, and the parts coming from licensed sources are explicitly stated as such.

- When implementing a method, a reference to the corresponding paper in the references page should be added.

## I  Time-series Classification Benchmark

Although the focus of `Torch-Uncertainty` has been on computer vision data, our library can be used for other application domains on the supported tasks. In this section, we leverage the `ClassificationRoutine` for Time-series classification with InceptionTime [42]. Specifically, we compare the following approaches on some of the UCR/UEA datasets [15]:

- **Baseline** a classic InceptionModel, the backbone for all other models.

- **Variational BNN** A BNN trained with the ELBO loss, sampling 16 models for evaluation.

- **MC Dropout** At test time, executes 10 forward passes of an InceptionTime model with a dropout rate of 0.2.

- **Ensembles** All ensembles have 4 subnetworks.

For evaluation, we consider the accuracy (**Acc**), the expected calibration error (**ECE**), the false positive rate at 95% (**FPR$_{95}$ (%)**), and additionally report the number of giga floating point operations (**FLOPS (G)**).

Table 10 summarizes the results obtainable in `Torch-Uncertainty` on this task.

## J  Text Classification Benchmark

The use of the library can be also be extended to tasks like Natural Language Understanding (NLP). In this section we fine-tune a `bert-base-uncased` [18] classifier initialized from the HuggingFace checkpoint on SST-2 [77] a sentiment analysis dataset and benchmark different baselines. Since there is no official test set, we use the validation set for testing, while setting aside part of the training set for validation.

**Tokenization** We use the `bert-base-uncased` tokenizer with `max_length`=128, truncation, and padding to max length. A deterministic split is applied: the first 3,000 rows of the GLUE [87] train split serve as validation; the remainder forms the training set. Evaluation is reported on the official GLUE validation split (872 labeled examples).

**Optimizer and schedule.**  We use AdamW as optimizer with decoupled weight decay (weight_decay = 0.01) and exclude bias and LayerNorm weights from decay. The learning rate

Table 10: **Time-series classification benchmark (averaged over three runs) on UCR/UEA datasets using an InceptionTime backbone.** All ensembles have 4 subnetworks. We highlight the best performance in **bold**.

| Method | Adiac | | | | Beef | | | |
|---|---|---|---|---|---|---|---|---|
| | Acc (%) | ECE (%) | $FPR_{95}$ (%) | FLOPS (G) | Acc (%) | ECE (%) | $FPR_{95}$ (%) | FLOPS (G) |
| Baseline | 76.34 | **5.84** | 34.39 | **2.17** | 75.00 | 21.08 | 70.83 | **5.81** |
| **Bayesian NNs** | | | | | | | | |
| Variational BNN (VI ELBO) | 78.01 | 6.94 | **29.82** | 34.81 | **77.78** | 20.86 | 70.83 | 92.96 |
| **Post-Hoc Methods** | | | | | | | | |
| MC Dropout | **100.00** | 6.03 | 53.51 | 15.82 | 70.83 | 18.70 | **63.89** | 58.10 |
| **Ensembles** | | | | | | | | |
| MIMO ($\rho = 0.5$) | 72.12 | 12.09 | 39.05 | 2.22 | 65.28 | 18.64 | 77.78 | 5.91 |
| BatchEnsemble | 73.27 | 11.99 | 47.49 | 8.70 | 65.28 | **15.80** | 76.39 | 23.24 |
| Packed-Ensembles ($\alpha = 2$) | 75.81 | 9.93 | 36.41 | 2.19 | 73.61 | 17.73 | 70.84 | 5.84 |
| Deep Ensembles | 78.28 | 6.71 | 41.34 | 8.70 | 66.67 | 16.80 | 81.94 | 23.24 |

| Method | CricketY | | | | CricketZ | | | |
|---|---|---|---|---|---|---|---|---|
| | Acc (%) | ECE (%) | $FPR_{95}$ (%) | FLOPS (G) | Acc (%) | ECE (%) | $FPR_{95}$ (%) | FLOPS (G) |
| Baseline | 87.28 | 4.15 | 83.10 | **3.71** | 88.18 | **3.64** | 56.51 | **3.71** |
| **Bayesian NNs** | | | | | | | | |
| Variational BNN (VI ELBO) | 87.00 | 7.15 | 81.62 | 59.34 | 88.09 | 5.65 | 60.39 | 59.34 |
| **Post-Hoc Methods** | | | | | | | | |
| MC Dropout | **87.56** | 5.50 | 85.61 | 37.09 | 88.37 | 7.80 | 53.19 | 37.09 |
| **Ensembles** | | | | | | | | |
| MIMO ($\rho = 0.5$) | 84.96 | 6.14 | 97.21 | 3.78 | 86.61 | 8.14 | 64.82 | 3.78 |
| BatchEnsemble | 85.79 | 7.70 | **69.73** | 14.83 | 87.17 | 9.21 | 49.58 | 14.83 |
| Packed-Ensembles ($\alpha = 2$) | 87.00 | 9.30 | 77.53 | 3.73 | **88.55** | 10.62 | **49.49** | 3.73 |
| Deep Ensembles | 85.24 | **3.82** | 80.22 | 14.83 | 87.26 | 7.95 | 54.20 | 14.83 |

| Method | Inline Skate | | | | Lightning7 | | | |
|---|---|---|---|---|---|---|---|---|
| | Acc (%) | ECE (%) | $FPR_{95}$ (%) | FLOPS (G) | Acc (%) | ECE (%) | $FPR_{95}$ (%) | FLOPS (G) |
| Baseline | 40.41 | 4.95 | 83.44 | **23.26** | **86.89** | 20.76 | 87.43 | **3.94** |
| **Bayesian NNs** | | | | | | | | |
| Variational BNN (VI ELBO) | 40.88 | 6.39 | 84.63 | 372.24 | 83.06 | **16.55** | 89.07 | 63.09 |
| **Post-Hoc Methods** | | | | | | | | |
| MC Dropout | 38.01 | 6.66 | **76.62** | 232.65 | 85.79 | 20.44 | **80.33** | 39.43 |
| **Ensembles** | | | | | | | | |
| MIMO ($\rho = 0.5$) | 26.18 | **3.68** | 89.18 | 23.68 | 83.61 | 17.35 | 86.88 | 4.01 |
| BatchEnsemble | 41.82 | 6.91 | 84.70 | 93.06 | 84.70 | 18.65 | 97.81 | 15.77 |
| Packed-Ensembles ($\alpha = 2$) | **43.02** | 7.57 | 79.16 | 23.40 | 85.25 | 23.64 | 80.87 | 3.97 |
| Deep Ensembles | 40.28 | 4.66 | 87.04 | 93.06 | 85.80 | 14.57 | 92.35 | 15.77 |

| Method | Olive Oil | | | | Two Patterns | | | |
|---|---|---|---|---|---|---|---|---|
| | Acc (%) | ECE (%) | $FPR_{95}$ (%) | FLOPS (G) | Acc (%) | ECE (%) | $FPR_{95}$ (%) | FLOPS (G) |
| Baseline | 77.78 | 29.00 | 83.33 | **7.05** | **100.00** | 0.41 | 51.84 | **1.58** |
| **Bayesian NNs** | | | | | | | | |
| Variational BNN (VI ELBO) | **85.18** | 22.63 | 81.48 | 112.74 | **100.00** | 0.25 | 63.23 | 25.32 |
| **Post-Hoc Methods** | | | | | | | | |
| MC Dropout | 81.48 | 18.82 | **88.89** | 70.46 | **100.00** | 0.16 | 70.69 | 15.82 |
| **Ensembles** | | | | | | | | |
| MIMO ($\rho = 0.5$) | 74.07 | 25.06 | **88.89** | 7.17 | **100.00** | 1.26 | **9.72** | 1.61 |
| BatchEnsemble | 70.37 | **18.61** | **88.89** | 28.18 | **100.00** | 2.15 | 78.05 | 6.33 |
| Packed-Ensembles ($\alpha = 2$) | 79.63 | 25.46 | **88.89** | 7.09 | **100.00** | **0.14** | 40.54 | 1.59 |
| Deep Ensembles | 72.22 | 25.35 | **88.89** | 28.18 | **100.00** | 2.15 | 78.05 | 6.33 |

is $\eta = 8 \times 10^{-6}$. The schedule is *per step*: linear warm-up over $10\%$ of the total training steps followed by linear decay to zero. Gradient clipping is applied with $\ell_2 = 1.0$.

**Early stopping and checkpoints.** Training runs for up to 7 epochs with early stopping on validation accuracy (patience $= 2$, $\Delta = 5 \times 10^{-4}$). We checkpoint every epoch and select the model with the best validation accuracy; final numbers are reported on the held-out test split.

**OOD evaluation.** We consider two out-of-distribution (OOD) settings: (i) *Near-OOD*, where the task is still sentiment analysis but the data comes from domains other than movie reviews. (ii) *Far-OOD*, where the task is different from sentiment analysis, following recent NLP OOD protocols [56][46].

All the splits are already defined in the library, a `datamodule` for the dataset SST2 is present, it automatically downloads train, test and OOD evaluation datasets. Evaluation is also pretty

Table 11: **Results on text classification.** Metrics evaluate accuracy, calibration performance and OOD detection. We highlight the best performance in **bold**

| Method | Heads | Acc↑ | Brier↓ | NLL↓ | ECE↓ | aECE↓ | OOD Average | | |
| --- | --- | --- | --- | --- | --- | --- | --- | --- | --- |
| | | | | | | | AUROC↑ | FPR95↓ | AUPR↑ |
| SINGLE | 12 | 92.55 | 0.12 | 0.27 | 0.05 | **0.04** | 70.16 | 70.62 | 81.93 |
| DEEP ENSEMBLES (D) | 12 | **93** | **0.11** | **0.24** | **0.04** | **0.04** | **74.81** | **62.69** | **84.9** |
| MC DROPOUT | 12 | 92.55 | 0.13 | 0.31 | 0.05 | **0.04** | 72.23 | 67.36 | 81.96 |

straightforward thanks to the `classification routine` and the different baseline codes already present also in the library.

**Results.** Table 11 reports ID accuracy, calibration, and OOD detection performance for different uncertainty methods. Deep Ensembles achieves the best overall performance and classical lightweight approaches such as MC Dropout improves only OOD detection.

*Note on Baselines:* for the Deep Ensembles baseline, we avoid training three completely independent BERT [18] models from scratch to reduce computational cost. Instead, we fine-tune a shared pre-trained BERT backbone and train three separate classifiers with different random seeds on top of it (we denote it as DEEP ENSEMBLES (D)).

