# OpenReview forum: "Torch-Uncertainty: Deep Learning Uncertainty Quantification"
_NeurIPS.cc/2025/Datasets_and_Benchmarks_Track — NeurIPS 2025 Datasets and Benchmarks Track spotlight_

### Official Review · Reviewer_Whxp · 2025-06-15

**Rating:** 5
**Confidence:** 4

**Summary:**

In this paper, the authors have developed Torch-Uncertainty, an open-source, modular, and domain-general PyTorch and Lightning-based library designed to standardise and facilitate the development, evaluation, and benchmarking of uncertainty quantification (UQ) methods in deep learning. They have implemented six categories of UQ methods, supported diverse tasks and metrics, and provided empirical results demonstrating the utility and comparative performance of UQ methods.

**Additional Feedback:**

1. In the Introduction section, when you compared and tried to distinguish your library from the existing ones, please reference Table 1 - to indicate to the reader that a detailed comparison is coming because this is what a reader wants to see first.

2. Where is the code for benchmarking different methods? Did I miss it? This can help users to quickly benchmark new methods.

**Dataset Code Accessibility:**

Yes

**Dataset Code Comments:**

The complete code is available at the given GitHub link.

**Ethical Considerations:**

No, there are no or only very minor ethics concerns

**Final Justification:**

The authors have addressed my concerns, and I have already recommended acceptance of the paper. Thanks!

**Limitations Weaknesses:**

1. Despite its modularity, the current implementation is restricted to four main task types, with a strong focus on computer vision. Broader applicability (e.g. NLP, time series, graphs) is not demonstrated.

2. The library does not cover methods from one category (Gaussian Process), and also does not cover certain methods, such as Diffision-based, Conformal Reg, and mean-Variance Est. While they justified not including Gaussian Process-based methods due to scalability but they did not justify why the rest methods were not included, which are available in the other.

**Strengths Contributions:**

1. The paper developed the Torch-Uncertainty library for uncertainty quantification in deep learning based on Pytorch and Lightning. This is a comprehensive library as it covers UQ methods from six categories (including deep ensembles, Bayesian NNs, conformal prediction, and post-hoc calibration), includes 21 metrics, and covers four tasks as classification, regression, segmentation, and pixel-wise regression across multiple data modalities. Additionally, this is developed as a modular and extensible library for ease of extension. While Lightning-UQ seems very close to the proposed library but the authors have clearly justified the advantages of their library, like more metrics and ability to integrate methods from different categories etc.

2. The paper clearly discusses the existing UQ libraries and methods and compares its coverage of methods in Table 1 of the paper, which shows the comprehensive and wider coverage of the proposed library. The proposed library also includes 21 metrics, much higher than the existing libraries. The authors also claim that their design of library is unique as it allows to integrated existing methods which is lacking in the existing libraries.

3. The paper discusses an important problem and highlights its significance for real-world deployment of deep learning models. Overall, the paper is well-written.

4. they provided empirical results comparing methods across multiple dimensions of robustness and calibration, showing that Deep Ensembles generally outperform others but may suffer in calibration without post-hoc correction. They also provided several tutorials and contribution guidelines for supporting researchers.

---

> ### Author Rebuttal · Authors · 2025-07-30
>
> Dear Reviewer Whxp,
>
> We thank the reviewer for their thorough review. We address below the limitations that the reviewer has raised.
>
> ### On domain applicability beyond vision:
>
> Coming from the computer vision field, we primarily focus on image-related tasks, models, and datasets, with which we are most familiar. However, with the addition of the classic regression pipeline, we added support for UCI datasets.
>
> For the rebuttal, we have added models and datasets for time series classification to demonstrate the application of Torch-Uncertainty for this task. This is only a first step for supporting tasks involving time series data. We provide the results of some InceptionTime model variants on several UCR/UEA classification datasets. As of v0.7.0, Torch-Uncertainty features a LightningDataModule and a PyTorch Dataset to handle any UCR/UEA classification datasets. In addition, we report the performance of InceptionTime model variants on 8 UCR/UEA classification datasets and their associated configuration files to reproduce this experiment. They can be found in the `experiments/classification/ucr-uea` folder.
>
> In particular, we report the Accuracy (Acc), the Expected Calibration Error (ECE), and the Giga number of floating-point operations (FLOPs). In addition, we engineered an Out-Of-Distribution (OOD) Detection task by considering one class of the dataset as OOD. While this method might change in the future, the idea is to showcase a complete pipeline to evaluate time series classifiers. Our primary focus here is to provide the tools for training and evaluating models on time series classification. We might tweak the configuration to better fit SOTA in the future. In any case, the configuration files allow users to control the hyperparameters of the experiments and easily modify this benchmark to their needs.
>
> For legibility concerns, we only present results on two datasets in this rebuttal. The remaining experiments can be found in our repository within the `experiments/classification/ucr-uea` folder. All ensembles consider four members, while MC Dropout runs 10 forward passes with a dropout rate of 20% and the BNNs use 16 samples. The results are the average between three models.
>
> **Adiac**
> | Method | **Acc (%)** | **ECE** | **FPR**$_{95}$ | **FLOPS (G)** |
> |--------|-------------|---------|----------------|---------------|
> | Baseline | 76.34 | 5.84 | 34.39 | 2.17 |
> | BNN | 78.01 | 6.94 | 29.82 | 34.81 |
> | MC Dropout | 100.00 | 6.03 | 53.51 | 15.82 |
> | MIMO ($\rho=0.5$) | 72.12 | 12.09 | 39.05 | 2.22 |
> | BatchEnsemble | 73.27 | 11.99 | 47.49 | 8.70 |
> | Packed-Ensembles ($\alpha=2$) | 75.81 | 9.93 | 36.41 | 2.19 |
> | Deep Ensembles | 78.28 | 6.71 | 41.34 | 8.70 |
>
> **CricketY**
> | Method | **Acc (%)** | **ECE** | **FPR**$_{95}$ | **FLOPS (G)** |
> |--------|-------------|---------|----------------|---------------|
> | Baseline | 87.28 | 4.15 | 83.10 | 3.71 |
> | BNN | 87.00 | 7.15 | 81.62 | 59.34 |
> | MC Dropout | 87.56 | 5.50 | 85.61 | 37.09 |
> | MIMO ($\rho=0.5$) | 84.96 | 6.14 | 97.21 | 3.78 |
> | BatchEnsemble | 85.79 | 7.70 | 69.73 | 14.83 |
> | Packed-Ensembles ($\alpha=2$) | 87.00 | 9.30 | 77.53 | 3.73 |
> | Deep Ensembles | 85.24 | 3.82 | 80.22 | 14.83 |
>
> ### NLP Baselines
>
> We plan to add models and datasets for document classification in NLP to demonstrate how Torch-Uncertainty can be applied to this task. Specifically, we plan to introduce a new text LightningDataModule for the SST-2 dataset, which includes train, validation, and test splits, as well as Near-OOD and Far-OOD loaders.
>
> The Near-OOD datasets will include IMDb, Yelp Polarity, and Amazon Polarity. The Far-OOD datasets will include AG News, 20 Newsgroups, TREC-QC, MNLI, RTE, and WMT16. These will be integrated into the library so that they work out of the box with Torch-Uncertainty's training and evaluation pipeline.
>
> ### On the absence of certain method families:
>
> We thank the reviewer for their feedback. We would like to explain our design choices:
>
> * Concerning diffusion models, uncertainty estimation in this area is still relatively new, especially for generative tasks. When we first developed Torch-Uncertainty, most existing methods for diffusion models were not very flexible or easy to use. Yet, we expect our library to be helpful in this domain, as shown in PUNC [1], which uses Torch-Uncertainty. We also plan to explore how to support better uncertainty estimation for diffusion models in future versions of the library.
> * Concerning conformal regression, while we started with conformal classification, we are actively working on applying the same approach to regression tasks. While we do not support it yet, we expect to add these methods shortly (issue #207).
>
> ### On the absence of the configuration files:
>
> Currently, the configuration files for the segmentation and regression benchmarks presented in the paper are included in the GitHub repository under the `experiments` folder. Moreover, this folder contains configuration files from a paper [2] that used an earlier version of the library to benchmark ensemble methods (Deep Ensembles, Packed-Ensembles, MIMO, BatchEnsemble, etc.) on image classification tasks.
>
> With this rebuttal, we also add configuration files for time series classification on 8 UCR/UEA datasets.
> The structure of this folder is unsatisfactory for us, as it is challenging to find experiments associated with our published papers (issue #190). Hence, we will refine it to be more user-friendly and allow for efficient result reproduction.
>
> ### Additional comment
>
> We will make sure that Table 1 is clearly put forward to guide the reader to a detailed comparison of Torch-Uncertainty with relevant libraries.
>
> ### References
>
> [1] Franchi, G., Belkhir, N., Trong, D. N., Xia, G., & Pilzer, A. (2025). Towards Understanding and Quantifying Uncertainty for Text-to-Image Generation. In CVPR25.
>
> [2] Laurent, O., Lafage, A., Tartaglione, E., Daniel, G., Martinez, J.M., Bursuc, A., & Franchi, G. (2023). Packed ensembles for efficient uncertainty estimation. In ICLR23.

---

### Official Review · Reviewer_t8yM · 2025-07-01

**Rating:** 5
**Confidence:** 4

**Summary:**

This paper introduces Torch-Uncertainty, a unified framework designed to streamline the workflow of uncertainty quantification (UQ) in deep learning. Existing UQ libraries often suffer from limited scope—focusing on specific types of uncertainty or narrow application domains such as regression—and frequently lack extensibility and computational efficiency. In contrast, the proposed framework aims to be comprehensive, extensible, and domain-agnostic, with a strong emphasis on evaluation. The primary contribution of this work is to provide a general-purpose, research-friendly library that facilitates rigorous and flexible experimentation in uncertainty quantification for deep learning models.

**Dataset Code Accessibility:**

Yes

**Dataset Code Comments:**

The code of the library is provided in the paper with detailed guidance on how to implement the library.

**Ethical Considerations:**

No, there are no or only very minor ethics concerns

**Final Justification:**

I thank the authors for their response and the additional results.
- I appreciate the explanation and additional experimental results on domains other than computer vision. I look forward to seeing the addition of these works in the updated version of the paper and the package.
- The results of the computation cost and the plan of adding the corresponding metrics in the package look good to me. I will accordingly increase my score.

**Limitations Weaknesses:**

1. The paper focuses on the design and efficiency of the library, which I believe are essential aspects of library development. However, in terms of domain coverage, the library could be further expanded by incorporating additional domains such as time-series data, graph-based models, and large language models (LLMs). Moreover, within the conformal prediction family, there is a growing body of work addressing challenges related to non-exchangeability in temporally structured tasks, which could also be considered for inclusion.

2. Some of the uncertainty quantification methods, such as the Bayesian-based methods, usually require a lot more computational cost than other uncertainty quantification methods or methods without uncertainty quantification. Thus, the absence of efficiency-related evaluation metrics in the library may limit its flexibility and hinder fair comparisons across different methods.

**Strengths Contributions:**

1. The library integrates a wide range of state-of-the-art uncertainty quantification methods and offers multiple evaluation metrics to support a comprehensive assessment of uncertainty across diverse application domains.

2. The library is designed with a rigorous and modular architecture, making it easy to extend. In addition to quantitative analysis, it also includes a visualization toolkit to further support and enhance uncertainty evaluation.

3. The paper's writing is good, and the overall structure is clear.

---

> ### Author Rebuttal · Authors · 2025-07-30
>
> Dear Reviewer t8yM,
>
> We thank the reviewer for their encouraging review. We address below the limitations that the reviewer has raised.
>
> ### Regarding the suggestion to expand domain coverage:
>
> We are working on expanding the scope of Torch-Uncertainty, despite limited resources. We completely agree that it's essential to support more tasks, such as time-series forecasting and large language models (LLMs). We had not planned to extend Torch Uncertainty to graphs until now. Still, we would be happy to collaborate with experts in this community to understand which directions would be most useful. Concerning conformal prediction, our next step is to add it for regression tasks, as we did for classification tasks.
> For this rebuttal, we have added new models and datasets for time-series classification and are working on NLP document classification. These examples show how Torch-Uncertainty can be used for tasks outside of computer vision, starting from v0.7.0 of the library.
>
> ### Time Series Classification
>
> We have added support for models and datasets for time series classification to demonstrate the application of Torch-Uncertainty for this task. This is only a first step for supporting tasks involving time series data. We provide the results of some InceptionTime model variants on several UCR/UEA classification datasets. As of v0.7.0, Torch-Uncertainty features a LightningDataModule and a PyTorch Dataset to handle any UCR/UEA classification datasets. In addition, we report the performance of InceptionTime model variants on 8 UCR/UEA classification datasets and their associated configuration files to reproduce this experiment. They can be found in the `experiments/classification/ucr-uea` folder in our repository.
>
> In particular, we report the Accuracy (Acc), the Expected Calibration Error (ECE), and the Giga number of floating-point operations (FLOPs). In addition, we engineered an Out-Of-Distribution (OOD) Detection task by considering one class of the dataset as OOD. While this method might change in the future, the idea is to showcase a complete pipeline to evaluate time series classifiers with respect to UQ. Our primary focus here is to provide the tools for training and evaluating models on time series classification. We might tweak the configuration to better fit SOTA in the future. In any case, the configuration files allow users to control the hyperparameters of the experiments and easily modify this benchmark to their needs.
>
> For legibility concerns, we only present results on two datasets in this rebuttal. The remaining experiments can be found in our repository within the ‘experiments’ folder. All ensembles consider four members, while MC Dropout runs 10 forward passes with a dropout rate of 20% and the BNNs use 16 samples. The results are the average between three models.
>
> **Adiac**
> | Method | **Acc (%)** | **ECE** | **FPR**$_{95}$ | **FLOPS (G)** |
> |--------|-------------|---------|----------------|---------------|
> | Baseline | 76.34 | 5.84 | 34.39 | 2.17 |
> | BNN | 78.01 | 6.94 | 29.82 | 34.81 |
> | MC Dropout | 100.00 | 6.03 | 53.51 | 15.82 |
> | MIMO ($\rho=0.5$) | 72.12 | 12.09 | 39.05 | 2.22 |
> | BatchEnsemble | 73.27 | 11.99 | 47.49 | 8.70 |
> | Packed-Ensembles ($\alpha=2$) | 75.81 | 9.93 | 36.41 | 2.19 |
> | Deep Ensembles | 78.28 | 6.71 | 41.34 | 8.70 |
>
> **CricketY**
> | Method | **Acc (%)** | **ECE** | **FPR**$_{95}$ | **FLOPS (G)** |
> |--------|-------------|---------|----------------|---------------|
> | Baseline | 87.28 | 4.15 | 83.10 | 3.71 |
> | BNN | 87.00 | 7.15 | 81.62 | 59.34 |
> | MC Dropout | 87.56 | 5.50 | 85.61 | 37.09 |
> | MIMO ($\rho=0.5$) | 84.96 | 6.14 | 97.21 | 3.78 |
> | BatchEnsemble | 85.79 | 7.70 | 69.73 | 14.83 |
> | Packed-Ensembles ($\alpha=2$) | 87.00 | 9.30 | 77.53 | 3.73 |
> | Deep Ensembles | 85.24 | 3.82 | 80.22 | 14.83 |
>
> ### NLP Baselines
>
> We plan to add models and datasets for document classification in NLP to demonstrate how Torch-Uncertainty can be applied to this task. Specifically, we plan to introduce a new text LightningDataModule for the SST-2 dataset, which includes train, validation, and test splits, as well as Near-OOD and Far-OOD loaders.
>
> The Near-OOD datasets will include IMDb, Yelp Polarity, and Amazon Polarity. The Far-OOD datasets will include AG News, 20 Newsgroups, TREC-QC, MNLI, RTE, and WMT16. These will be integrated into the library so that they work out of the box with Torch-Uncertainty's training and evaluation pipeline.
>
> ### Regarding efficiency-related evaluation metrics:
>
> We agree with the Reviewer t8yM concerning the measurement of the computational cost linked to the different baselines we implement. With Torch-Uncertainty v0.7.0, we add to all our routines (Classification, Regression, Segmentation, PixelRegression) the computation of the number of the model’s parameters and the number of floating-point operations in a forward pass. We are also considering adding a flag to compute the throughput of a model at inference. However, this value will be subject to variability depending on the hardware accelerator used, among others.

---

### Official Review · Reviewer_2GGN · 2025-07-03

**Rating:** 5
**Confidence:** 4

**Summary:**

The authors present a pytorch framework for uncertainty quantification made for evaluating reliability of model outputs, and hence making decisions after taking uncertainty soundly into account. By being built on top of Pytorch-lightning framework, the package makes it modular to combine multiple uncertainty quantification methods, even those which are post-hoc. It also provides an extensive and compreshensive coverage over UQ metrics divided over multiple tasks such as classification, out-of-distribution detection, selective classification, calibration, diversity, regression/depth prediction, and segmentation providing a one stop place for all kind of UQ needs and metrics. OOD evaluation is missing from many other popular and competing packages, which is an important task in industry especially. The package does not provide support for GP based models but covers frequentist conformal predictions method for UQ.

**Dataset Code Accessibility:**

Yes

**Dataset Code Comments:**

All the code is online including models, datasets and scripts.

**Ethical Comments:**

No comments.

**Ethical Considerations:**

No, there are no or only very minor ethics concerns

**Final Justification:**

I think, I will stick with my score after following the discussion.

**Limitations Weaknesses:**

- No GP support and no support for Diffusion models.
- As correctly identified by the authors as well, the modular approach means that practitioners might try to combine methods which are not compatible or even from the same domain as each other. Does the package report these kind of failure modes to the end users.

**Strengths Contributions:**

- The package is modular built on top of Pytorch lightning
- Despite there being many similar packages available online, this offers something different to users.
- Implementation and possibility to combine multiple UQ methods, even those which are post hoc.
- The paper is well written, documented and provides a lot of references to a new comer.
- The paper thoroughly discusses its advantages, shortcomings and weaknesses.

---

> ### Author Rebuttal · Authors · 2025-07-30
>
> Dear Reviewer 2GGN,
> We thank the reviewer for their detailed and positive feedback. We address below the limitations that the reviewer has raised.
> ### Regarding the lack of Gaussian Process (GP) and diffusion model support:
> Currently, we don’t feature Gaussian Processes (GP) within Torch-Uncertainty. As explained in the paper, our primary focus is to study UQ in Deep Neural Networks, including relatively large architectures, while GPs might not scale very easily. However, we agree that GPs are a paramount subclass of models in uncertainty and plan to support some of their variations shortly. We have added an issue (#206) to keep track of and mitigate this limitation.
> Concerning diffusion models, uncertainty estimation in this area is still relatively new, especially for generative tasks. When we first developed Torch-Uncertainty, most existing methods for diffusion models were not very flexible or easy to use. Yet, we expect our library to be helpful in this domain, as shown in PUNC [1], which uses Torch-Uncertainty. We also plan to explore how to support better uncertainty estimation for diffusion models in future versions of the library.
> Regarding the expansion of supported tasks, please note that we have already increased the scope of the library by adding support for time-series classification in Torch-Uncertainty v0.7.0, and we are currently working on NLP.
> ### On the risk of combining incompatible methods due to modularity:
>
> We thank the reviewer for this question. First, let us recall that, while flexible, our pipeline imposes some constraints, as depicted in Figure 2 of the main paper. For instance, only one post-hoc method is available at a time (e.g., Temperature scaling, conformal predictions, etc.). However, some unusual combinations could still be selected, such as a Bayesian Neural Network followed by a Laplace approximation. While we believe that unorthodox mixes could be interesting for researchers, we agree that some of them might not be desirable, especially for newcomers. We will look into identifying those and implementing a warning system, while improving our tutorials to guide users towards the more classical solutions.
>
> ### References:
>
> [1] Franchi, G., Belkhir, N., Trong, D. N., Xia, G., & Pilzer, A. (2025). Towards Understanding and Quantifying Uncertainty for Text-to-Image Generation. In CVPR25.

---

> > ### Comment · Reviewer_2GGN · 2025-08-05
> > **keeping the score same as before**
> >
> > I have read the author's response to my review and other reviewers and still maintain my score for this submission. I thank the authors for replying to my comments.

---

### Official Review · Reviewer_wkte · 2025-07-07

**Rating:** 5
**Confidence:** 3

**Summary:**

This paper proposed a uncertainty estimation tool for deep learning for classification, segmentation, and regression. The tool has a high code quality standard, and implements SOTA  uncertainty quantification methods for deep learning with good modular design. Besides, based on proposed tool, this work also provided one benchmark on image classification and the other benchmark on semantic segmentation.

**Dataset Code Accessibility:**

Yes

**Ethical Considerations:**

No, there are no or only very minor ethics concerns

**Final Justification:**

I think this work is very soild, and if the plan can be achieved in the future, this work will have a high-level impact. Hence, I keep my positive rating.

**Limitations Weaknesses:**

1. The baselines for classification benchmark are a little lacking.
2. Could this tool support the NLP domain and more large model like LLM?

**Strengths Contributions:**

1. The documents for tool usage  are very clear.
2. The tool support different modalities such as segmentation and tabular datasets.
3. One benchmark on image classification and the other benchmark on semantic segmentation are provided.
4. The usage of this tool for training and evaluation is easy, and the multiple metrics for model selection are nice.

---

> ### Author Rebuttal · Authors · 2025-07-30
>
> Dear Reviewer wkte,
>
> We thank the reviewer for their constructive review. We address below the limitations that the reviewer has raised.
>
> ### On the restricted number of classification baselines:
>
> We recognize that the current classification benchmark includes only a few baseline models. Our initial goal was to highlight baselines using Vision Transformers (ViTs), where uncertainty quantification is still underexplored. We trained all models from scratch, including a fully pre-trained Deep Ensemble of ViTs, as well as MIMO and Packed Ensembles. We plan to release the code shortly, as well as the checkpoints on our HuggingFace account. We will let the reviewer know if we manage to move forward on this aspect during the discussion period.
> In another paper [1], based on an early version of the library, we provided experiments with  ResNet variants. We could re-run them with our latest code and compile all the experiments in the final version of this paper if the reviewer finds it relevant.
>
> ### Regarding the suggestion to support the NLP domain and large models like LLMs:
>
> Our objective is to expand the library's scope to support more tasks, including modern models such as vision-language models (VLM), large-language models, and multimodal large-language models (mLLM). However, we constantly need to balance our efforts between improving the usability of Torch-Uncertainty, maintaining it, and implementing new tasks and features, given our limited resources. These constraints have so far hindered the implementation of such models in our library, but we will keep the reviewer’s suggestion in mind for our subsequent releases.
> We plan to add support for NLP document classification in the coming months. In that regard, we have conducted experiments on NLP encoders that are yet to be added to our main branch. Contributions from the community to extend to more LLM tasks are also welcome.
>
> ### NLP Baselines:
>
> We plan to add models and datasets for document classification in NLP to demonstrate how Torch-Uncertainty can be applied to this task. Specifically, we plan to introduce a new text LightningDataModule for the SST-2 dataset, which includes train, validation, and test splits, as well as Near-OOD and Far-OOD loaders.
> The Near-OOD datasets will include IMDb, Yelp Polarity, and Amazon Polarity. The Far-OOD datasets will include AG News, 20 Newsgroups, TREC-QC, MNLI, RTE, and WMT16. These will be integrated into the library so that they work out of the box with Torch-Uncertainty's training and evaluation pipeline.
>
> ### References:
>
> [1] Laurent, O., Lafage, A., Tartaglione, E., Daniel, G., Martinez, J.M., Bursuc, A., & Franchi, G. (2023). Packed ensembles for efficient uncertainty estimation. In ICLR23.

---

> > ### Comment · Reviewer_wkte · 2025-08-06
> >
> > Thanks for the response. I think if the plan can be  achieved in the future, this work will have a high-level impact.

---

### Comment · Area_Chair_HXzg · 2025-08-01

Dear Reviewers,

Thank you for providing reviews. Now that the authors have had an opportunity to respond to the reviews, can you please carefully read all the other reviews and the author responses, acknowledge that you have read the author response to your questions/concerns, and check whether you want to raise any further questions/concerns at your early convenience, so that there is time for back and forth discussion with the authors within the time window? Finally, you should think about whether the other reviews and the author responses justify a change to your score (in either direction).

 Thank you very much for your time, effort, and contribution to the organization of NeurIPS 2025.

---

### Note · Authors · 2025-08-15

Dear Area Chair, dear Reviewers,

We would like to thank the reviewers once again for their evaluation of Torch-Uncertainty and its paper, as well as for their valuable remarks, which will help us improve the library. We have noted down all the limitations raised by the reviewers, already implemented and released some mitigations, such as the performance metrics, a time-series classification task, and will address the remaining ones in the near future.

We will continue to strive towards making uncertainty quantification more accessible to both researchers and deep learning practitioners.

---

### Decision · Program_Chairs · 2025-09-18

**Decision:**

Accept (spotlight)

**Comment:**

Torch-Uncertainty is an open-source, modular, and domain-general PyTorch and Lightning-based library designed to standardise and facilitate the development, evaluation, and benchmarking of uncertainty quantification (UQ) methods in deep learning. It implements six categories of UQ methods, supports diverse tasks and metrics, and provides empirical results demonstrating the utility and comparative performance of UQ methods.

### Strengths

- The code is modular, easy to use, and provides good coverage of modern UQ techniques and metrics.
- The paper is well-written and describes the package and the current state of neural net UQ well.

### Weaknesses

 - The code is more targeted towards computer vision applications, although the authors do add in time series during the rebuttal period.
- There is not yet support for conformal methods
- The reviewers also point out a lack of Gaussian process and diffusion methods, but the former is arguably not a neural network method, and UQ for generative diffusion models is still very new.

### Recommendation

This is a useful package and review of the current state of neural network UQ that will be useful for practitioners interested in UQ for a variety of applications. This paper stands out for its coverage of both techniques and metrics and the quality of the code package, which will make it generally useful for researchers interested in implementing new UQ methods and using current ones. Accept (spotlight).